# Sponge City Construction and Urban Economic Sustainable Development: An Ecological Philosophical Perspective

**DOI:** 10.3390/ijerph20031694

**Published:** 2023-01-17

**Authors:** Jing Ma, Dan Liu, Zhengwen Wang

**Affiliations:** 1Beijing Technology and Business University, Beijing 100048, China; 2School of Economics and Management, Wuhan University, Wuhan 430072, China; 3National Institute of Insurance Development, Wuhan University, Ningbo 315100, China

**Keywords:** ecological philosophy, sponge city construction, sustainable urban economic development

## Abstract

The Party’s 14th Five-Year-Plan and the 2035 Visionary Goals point out that green and sustainable development is the direction of China’s road in the present age, and provide a theoretical basis for further improvement of ecological civilization. “Sponge city” is a new type of urban construction idea in China; moving from pilot to vigorous implementation at present, it is an important element of China’s promotion of green development and development of ecological civilization. At present, a number of sponge city pilot projects have been built in China, and evaluation of their effects is already a matter of urgency. The overall planning of China’s current policies in sponge city construction and the specific analyses conducted by experts from both subjective and objective aspects have not been able to completely fill the gap in this regard, thus making it particularly urgent to conduct in-depth studies. Based on this, this paper discusses the performance assessment of sponge cities in China on the basis of the prediction and analysis of the development trend of sponge cities in China. In the performance assessment system, the correctness and timeliness of the index system should be considered in terms of practical effects; in the city performance assessment, the ideas of new city development such as low-carbon cities and smart cities should be combined to build a comprehensive and multi-perspective intelligent assessment system, so as to provide a strong boost to promote the development of city construction and its evaluative research. Firstly, a system-dynamic model is applied to sort out and combine its internal operation mechanism, and a set of evaluation systems based on the ecological philosophical perspective of the sponge city and urban sustainable development performance is established. Second, to improve the accuracy of the research results, parallelism tests and robustness analysis were conducted on this performance index evaluation system. The study’s results show that sponge city construction has achieved good results in sustainable urban development and has contributed to future development.

## 1. Introduction

In the second half of the twentieth century, with the rapid development of China’s urban economy and the rapid expansion of urban scale, the traditional concept of urban architecture formed since the Industrial Revolution has been severely impacted, especially the problem of urban water pollution, which has seriously affected the sustainable development of many cities in the world [1]. In recent decades, China’s urban construction has made remarkable achievements, but the traditional way of building urban stormwater systems has severely restricted the development of cities and caused serious water pollution and water shortages. Therefore, how to fundamentally solve the water problems in cities has become an urgent problem at present [2]. On this basis, the concept of “sponge city” has been proposed to provide a new way to solve the flooding problem. In terms of ecological benefit indicators, the construction of sponge cities is mainly to promote the improvement of the urban ecological environment, which is the material factor that can meet people’s needs for social production and the life that it contains [3]. Sponge city construction can mitigate natural disasters caused by rainfall, for example; in the rainy season, flooding and waterlogging occur more often in southern cities, while the construction of sponge cities can effectively reduce the urban heat island effect, control urban water pollution, reduce dust and noise in urban air, and provide sufficient space for the development of cities in a tropical monsoon climate [4].

The sponge city construction work in China is based on Marx’s ecological philosophy, which has been enriched and developed in practice [5]. The construction of sponge cities has led to an increase in the area of urban green space, and its improvement of the ecological environment has certain implications [6]. In sponge city construction, the collected rainwater can be used for road cleaning, greening irrigation, and residents’ daily water supply; in the new urban wastewater treatment facilities, the treated water can also be used for road cleaning, greening irrigation, and residents’ daily water use [7,8]. The construction of the sponge city will produce certain social effects, such as expanding the green space and ecological environment of the city and increasing new employment opportunities. In the construction of sponge city, the concept of green environment protection is greatly promoted and a low-carbon lifestyle is integrated with the city. The collected rainwater can be used for road cleaning, greening and irrigation, and domestic water for residents, thus improving the ecological quality of the city [9,10,11]. Therefore, after the construction of sponge cities, urban green areas have been greatly improved and the ecological environment of cities has been greatly enriched [12]. Based on this, in the construction of the sponge city, the pollution status of the river should be taken into account and low-impact measures should be adopted, as well as a more scientific approach to greening, so that the function of the river system can be maximized. Additionally, a series of rainfall infiltration measures are adopted, which on the one hand replenish the groundwater and rich water sources; on the other hand, by truncating the infiltration of rainwater, the convergence of rainwater is reduced and all the rainwater is concentrated within the rainfall measures, which greatly reduces the runoff and controls the waterlogging in the city [13,14,15].

To sum up, with the accelerated urbanization process, the construction of sponge cities in China is also booming. And sponge cities must be explored from the perspective of ecological philosophy in solving floods, promoting economic development and building a harmonious urban ecological environment with humans and water. In this paper, we philosophize about the problems in the construction of sponge cities in China from the viewpoint of ecological philosophy, analyze the causes, and propose corresponding countermeasures, which will help people change the concept of urban construction from a deeper level, so as to promote the construction of sponge cities in China and thus achieve the purpose of sustainable development.

## 2. Literature Review

Meng, Li, and Tang (2021) proposed in their study that, with the accelerated urbanization, the urban area is expanding, but the increasing pressure on urban drainage facilities for flood control and drainage is due to the widespread use of hard surfaces on urban roads, the dramatic increase in the proportion of green land per capita, and the dramatic decrease in rainwater infiltration and increase in runoff volume [16]. In the case of excessive rainfall during the flood season, some areas in China experience large amounts of waterlogging, which causes serious flooding [17]. This not only has an impact on the daily life and production of residents, but also has a negative impact on the development of cities [18]. In order to reduce flooding and improve the flood control capacity of urban areas, it is necessary not only to improve the flood control standards of drainage facilities, but also to apply the concept of “sponge city” to urban development and construction to enhance the flood control capacity of urban drainage systems and improve the efficiency of urban water resources utilization in order to protect the urban ecological environment and ensure the normal operation of urban functions [19,20,21]. Tang, Jiang, and Zheng (2020) stated in their study that in sponge city construction, it is important to consider all aspects and reform priorities to provide legal basis and economic support for its development, and it should also make full use of resources in government decision-making, law, design engineering systems, and public participation to play an important role in sponge cities [22,23]. Yang and Sang (2020) argued that the construction of sponge cities is important for the green and ecological development of cities [24]. The traditional urban building roads are all hardened roads, and the planning and design ideas of “fast exclusion” and “end concentration” are dominant, which are prone to “flooding in every rain” and “drought and flooding”. The problem of “flooding and drought” has seriously affected the green ecological development of the city [25,26,27]. The main planning and design idea of sponge city is “slow drainage and slow release”, or “source dispersion”, using urban green space, roads, water systems, etc. to absorb, store and slowly release rainwater, minimizing urban development and construction’s effects on the original natural hydrological characteristics of the city and water The “sponge body” scale and quality will be improved so that the roof of the city will “green up” and the urban green space will “sink down”. Urban green areas “sink” [28,29,30].

Wang, Wu, and Chiles (2022) showed in their study that urban construction has significantly contributed to the development of China [31]. By establishing sponge cities, rainwater flood management can be effectively carried out to make them better adapt to the environment and cope with natural disasters caused by rainwater, providing a strong guarantee for the sustainable development of cities [32,33,34]. The sponge city integrates flood control and drainage, and it can effectively relieve the pressure of urban flood control and drainage, prevent floods and other natural disasters, and lay a good foundation for the stable development of the city [35,36,37]. By restoring and protecting urban wetland vegetation, it can further green the city, reduce the heat island effect, and improve the quality of life of its residents [38]. By implementing “sponge cities”, the workload of existing urban drainage facilities can be significantly reduced, thus reducing the investment in urban infrastructure [39,40,41]. In sponge cities, there will also be better integration of existing gardens and green spaces within the sponge cities, which will not only reduce construction costs, but also reduce costs related to water pollution, thus reducing urban losses due to flooding [42,43,44,45]. Currently, China has started to pay attention to the construction of sponge cities and has contributed to the development of cities to some extent [46]. We need to put the benefits in the first place, but also develop the idea at a deeper level to achieve comprehensive utilization [47], thereby further realizing the organic unity of water culture and water landscape, promoting the integration of water culture, and promoting the economic development of the city [48].

Kiconco, Ruhinda, Kyobutungi (2017) argued that the problems of water resources are more complex, such as: flood control and drainage, scientific use of water resources, etc., as well as landscape water, water pollution, etc., and many factors are involved, so in the process of establishing an integrated rainwater management system, it is necessary to take multiple objectives as a starting point, combine the actual situation, and take into account the process control and source control [49,50,51,52]. In addition, the process of construction involves hydrology, meteorology, water conservation, environmental considerations, etc., which are the responsibility of municipalities [53]. The environmental protection, municipalities, and landscape are an integral part of the construction of the rainwater management system of sponge cities [54,55]. The key to the “sponge city” is to ensure urban green space and water absorption, infiltration, and water purification capabilities, not only to pay attention to the functions of urban green space and roads, but also to carry out scientific and reasonable planning of urban land layout and vertical design, and efficiently connect low-impact development rainwater facilities, urban rainwater irrigation systems, and rainwater runoff systems, so that the synergy of “green” infrastructure and “gray” infrastructure can be maximized [56,57].

Sustainable development has the following meanings: (1) natural properties: “maintaining and enhancing the productive and regenerative capacity of ecosystems”; (2) social properties: “improving the quality of human existence, within the carrying capacity of the ecological environment”; (3) economic properties. “to guarantee the improvement of the well-being of the current generation without reducing the well-being of future generations”; (4) technical nature: “to transform cleaner and more efficient technologies into the maximum “zero emission” or “confined“ production processes to minimize the consumption of energy and other resources [58,59]. By defining these concepts, we can have a general idea of the development of sponge cities. The rapid development of productivity has, on the one hand, raised the standard of living of human beings and promoted the development of society; on the other hand, it has also led to a great desire to expand the scope of activities to areas that had never been covered before [60,61]. Especially in the rapid urbanization process, people’s living environment is deteriorating due to the misuse of resources, and people’s living areas are gradually eroded by “gray”. Sponge cities, with sustainable development as the core, are an important element in promoting urban development and construction [62]. The traditional urban construction problems highlighted, the need for ecological environmental protection, and the increasing severity of urban flooding make the study of sponge cities particularly important, and the promotion of a high-quality road to green development is also a consensus. With the full implementation of China’s sustainable development strategy, the 16th National People’s Congress proposed “accelerating the construction of an international metropolis”; after the 18th National Congress, the idea of “new urbanization” has become increasingly prominent, emphasizing people-oriented, intensive and efficient, and green and smart urbanization [63]. The National New Type Urbanization Plan (2014–2020) clearly proposes to combine the idea of ecological civilization with the urbanization process, “vigorously promote green development, circular development, and low-carbon development, and promote green and low-carbon production and lifestyles and urban construction and operation modes” [64]. The 19th National Congress proposed to firmly establish the concept of socialist ecological civilization, create a new situation of harmonious coexistence between human beings and nature, and make due contributions to the realization of ecological civilization. At the same time, the new development concept also highlights the coordinated development of economy, society, and environment to achieve the harmonious development of man and society, and man and nature [65].

The “sponge city” refers to the transformation of nature to provide a suitable living environment for people, so as to achieve their material and psychological needs. However, the traditional non-scientific urban construction has caused changes in surface runoff and increased the load on the underground pipe network. Urban flooding, on the surface, appears to be due to the inability of urban drainage systems to adapt to urban development, but in reality, its root cause is water shortage, water security, and water environment problems [66,67]. This is mainly because in traditional municipal engineering, engineering drainage facilities, drainage engineering planning, and flood control planning are relatively lagging behind, and the awareness of reusing rainwater resources is poor. While urban construction meets people’s basic housing needs, there is also a waste of resources and a large amount of construction waste that damages the environment [68]. In practice, it has been recognized that the traditional approach to urban development is undesirable and unsustainable, so it is important to explore the path of “sustainable” urban development. On this basis, this paper proposes a performance assessment index for ecological, economic, and social aspects of cities based on sustainable development, and establishes corresponding performance assessment indexes to contribute to the development of ecological, economic, and social aspects of cities.

## 3. Research Design

### 3.1. Data Sources

The article selects 34 cities in the eastern region as the research object, and the original data mainly comes from the China Statistical Yearbook 2020, the Statistical Bulletin of National Economic and Social Development, the Water Resources Bulletin, and the Statistical Yearbook of Urban Construction of each city. In this paper, by referring to the relevant indicators of sustainable urban development, low-carbon cities, green cities, etc., and also drawing on the research results and survey reports of previous topics, we start from the objectives and significance of sponge city construction, select the corresponding evaluation system from them for the principles of evaluation indicators selected from the above literature, and change and reorganize to determine the final evaluation system, Table 1 shows the performance evaluation system of sponge city construction.

### 3.2. Evaluation Indicators

By reviewing the literature this paper refers to the theoretical research results of Duke T, Stiger-Pouvreau V, Corazza L, et al. [66,67,68]. The following indicators were selected for the subsequent empirical analysis of this paper, the details of which are as follows.

#### 3.2.1. Driving Force Indicators

The sponge city is a complex system, and the main reasons for this change are social-level reasons, such as the accelerated rate of social development and increased urban expansion, with relatively small ecological and economic drivers. Among the social drivers, three main representative indicators were selected, namely, population size, urbanization rate and GDP.

(1) Population size. Population size refers to the number of living individuals in certain conditions of time and place. The specific number of the population, or the total population, is usually related to a specific spatial and temporal condition. Population size that is divorced from the prerequisites is also meaningless. Population size has three characteristics: first, a defined time; second, a defined place; and third, a defined type (including permanent or temporary residence). (2) Urbanization. The urbanization rate (also known as the urbanization rate) is a measure of the level of urbanization, usually using demographic methods, that is, the proportion of the rural population in the total population. (3) Gross domestic product (GDP). GDP is the result of the productive activity of all resident units of a country (or region) over a certain period of time and is usually considered the best indicator of a country’s economic situation. GDP as an important comprehensive statistical indicator reflects the economic strength and market size of the country within the national economic accounting system.

#### 3.2.2. Pressure Indicators

Social development, urban expansion, and population changes have influenced the urban system, bringing pressure to cities in three aspects: ecological, economic, and social. To reflect this pressure, eight indicators, such as urban construction land area, are selected in this paper to reflect the pressure on these three aspects.

The ecological aspect is mainly reflected in three places: (1) Urban construction land. The area of urban construction land refers to the area within the urban area that has been developed and constructed, and where municipal facilities and public facilities are basically available. In the central urban area, it mainly consists of some concentrated areas and some scattered areas where municipal facilities and public facilities are available. (2) The area of green space per capita. According to the “Urban Green Space Classification Standard” CJJ/T85-2002, urban green space is divided into five categories: park green space, production green space, protective green space, subsidiary green space and other green space. Park green space per capita = park green space/number of urban residents. (3) Sewage. Sewage mainly includes domestic sewage, industrial wastewater, initial rainwater, etc., all of which contain a large amount of organic matter, and all have lost their original uses.

The economic aspect is reflected in two main places: (1) The share of government revenue in total revenue is the proportion of GDP of all total capital required by the government to perform its functions, implement public policies, and provide public goods and services. (2) The amount of government investment in sponge city construction refers to the funds invested in the construction of sponge city led by the government.

The social aspect is mainly reflected in three places: (1) Water resources per capita refers to the total amount of water resources owned by the population according to a certain area (watershed) at a certain time. (2) Water for the ecological environment refers to the minimum amount of water needed to restore, build or maintain the quality of the existing ecological environment. (3) The total amount of water saved refers to the water resources saved by technical means or recycling methods compared with the original water use.

#### 3.2.3. Status Indicators

Under pressure from three dimensions: ecological, economic and social, the urban system has emerged in a very different situation than before. Seven assessment indicators, such as sponge areas, are used as descriptive features.

There are five main indicators of ecological characteristics: (1) Sponge area is the sum of man-made and natural areas in the city; they have the role of “sponge”. (2) Natural lake maintenance rate refers to a certain period of time, ensuring that the city’s natural lakes occupy a relatively good proportion of the area. (3) Ecological shoreline ratio refers to the proportion of ecological shoreline area to the whole shoreline area within the urban river section. (4) The total green area of the built-up area is the ratio of individual green areas to the total land area within the built-up area of the city. Greening coverage ratio = (greening area/land) × 100%. (5) Wastewater treatment rate refers to the ratio of the amount of domestic and industrial wastewater in the sewage after sewage treatment. Wastewater treatment rate = wastewater treatment volume/total sewage volume × 100%.

There is one main indicator of economic characteristics: the construction of sponge cities. The volume of sponge city construction is the number of sponge cities that have been built in our cities.

There is one main indicator of social characteristics: the price of residential water. The price of residential water refers to the combined price of tap water and sewerage charges.

#### 3.2.4. Impact Indicators

The change in the condition of the urban system has both positive and negative socio-economic impacts, as well as ecological, economic and social dimensions. In this paper, five indicators have been chosen to represent the impact of the change through the rate of compliance with the quality standards of surface water.

The ecological dimension mainly contains one indicator: the rate of surface water quality compliance. The rate of surface water quality compliance is the total number of water bodies with use functions, including rivers, lakes, reservoirs, etc., that reach the corresponding grade according to GB3838-02.

The economic dimension mainly contains one indicator: the capital investment in the sponge city. Sponge city investment refers to the resources invested by social capital in the construction of sponge city during the construction process.

The social dimension contains three main indicators. (1) The volume of runoff pollution control in the watershed is the volume of stormwater that needs to be treated for the purpose of achieving runoff pollution control and ensuring water quality. In the U.S. Application of Urban BMPs, the water quality control volume is expressed through the following equation:(1)WQV=10HϕF
where *WQV*—water quality control volume required, m^3^; *H*—design rainfall, mm; ϕ—runoff coefficient; *F*—catchment area, hm^2^. (1) Total annual runoff control rate refers to the use of natural and human-enhanced infiltration, detention and purification to regulate the precipitation in urban construction areas, so as to control the ratio of average annual precipitation to total annual precipitation. (2) Total annual SS removal rate is the removal rate of total suspended turbidity (SS) throughout the year, usually 40% to 60%. Suspended matter in municipal wastewater is one category of pollutant index, which has a certain correlation with other pollutant indexes, usually with SS as the index. Removal rate of total annual SS = total annual flow control rate × average removal rate of SS by low-impact equipment. The total annual SS removal rate for an urban or development area can be obtained by a weighted average of the annual runoff volume (annual precipitation × combined rainfall volume−runoff factor × catchment area).

#### 3.2.5. Reaction Index

When a city is affected by the above-mentioned positive or negative impacts, its corresponding response mechanism will include ecological, economic and social aspects in order to maintain its stability and sustainable development. Based on this, four assessment indicators, such as the ecological benefits of the project, were selected to reflect the response process.

Among the ecological factors are mainly the ecological benefit of the project, which refers to the resources that people use and the impact on the environment in the process of implementing the sponge city, and the ecological and economic perspective of the environment, as well as the optimal configuration of the two, in carrying out economic or other activities to minimize the use of resources and damage to the ecological environment in order to produce economic benefits.

Among the economic factors are mainly the economic benefits of the project, which refers to the process of sponge city construction, through trade with outside goods and labor, so as to achieve maximum productivity; that is, under the same labor consumption, to achieve greater production benefits.

There are two main social factors: (1) Engineering social benefits refers to the social benefits that can be generated by the construction and operation of sponge cities. The construction and operation of a specific project must, to a greater or lesser extent, cause a certain positive or negative impact on society. (2) The number of relevant policies promulgated refers to the number of policies, regulations, standards, etc. on sponge cities published by relevant departments.

### 3.3. Model Design

The comparison of sustainable development before and after the pilot sponge city is used to evaluate the effect of its implementation. On this basis, the actual situation in China is taken into account, taking into account the fact that natural conditions such as geographic location, temperature and precipitation have a certain influence on the development of the city, and also have a certain negative impact on the development of the city. Therefore, the sustainable development of cities cannot be attributed simply to the construction of “sponge cities”, and conversely, the construction of sponge cities may not promote the green economic development of cities. On this basis, a comparative study was conducted on the sustainable economic development of Chinese cities using the double DID model, and an empirical study was conducted on its contribution to the economic development of Chinese cities. The model is set up as shown below:(2)$$Y_{it}=a_{0}+a_{1}datacity_{it}+\beta control_{it}+v_{i}+\mu_{t}+\varepsilon_{it}$$

In the equation, $Y_{it}$ is the explanatory variable, *i* represents the city, *t* represents the time. $v_{i}$ is represents a fixed urban effect; $\mu_{t}$ is represents a fixed year; *datacity_it_* is represents the interaction term between sponge city and construction time, taking a value of 1 for both dummy variables of sponge city and construction time, and 0 in other conditions. Assessment factor is the indicator that needs special attention, as it represents the economic development of sponge city, greater $a_{1}$ > 0, indicating that it has a positive contribution to the sustainable development of the economy, $a_{1}$ = 0, indicating that it does not have a negative impact on the sustainable development of the economy, and $a_{1}$ < 0, indicating that the construction of sponge city will have a negative impact on the sustainable economic development of the city. The controlit is a control variable, referring to the economic agglomeration, technological innovation, education expenditure, and other influence factors. 

## 4. The Empirical Analysis

### 4.1. Analysis of Baseline Model Regression Results

Sustainable urban development is reflected in three aspects: economic, social and ecological. Therefore, to achieve the development of a city, it is necessary to focus not only on economic development, but also on the quality of development. The goal of the construction of a sponge city is to improve the quality of life of the residents, and its existence will have a certain impact on the economic development and environmental quality of the city. Based on the established base model, GDP per capita and industrial SO_2_ emissions were taken as explanatory variables and empirically tested with a regression model. The detailed results are shown in Table 2, after adding control factors to the model, the estimated coefficient of sponge city construction on GDP per capita in China was positive and around 1%, while the estimated coefficient on SO_2_ was significantly negative. The results show that sponge city construction can achieve both economic development and ecological environment improvement. The analysis of sponge city can make the economic development of the city and the improvement of environmental quality achieve a win-win situation.

This paper also examines total factor productivity and estimates it in Table 3. Model (2) is a regression that incorporates technological innovation, industrial upgrading, and infrastructure construction into model (1). The estimated coefficients of sponge city construction in both model (1) and (2) are positive and above 1%, which indicates that there is a large positive correlation between sponge city construction and total factor productivity. Technical efficiency refers to the difference between actual and optimal output, and technical progress refers to the change in the level of technology. The ml-ebm indicators were divided into technical efficiency indicators and technical progress indicators, and model (3) and model (4) further investigated the differences between the two indicators. It was found that the regression coefficient with technological efficiency as the explanatory variable was small and insignificant. The study with technological progress as the explanatory variable, on the other hand, obtained a positive coefficient value and tested the significance of the 1% test. It shows that the construction of sponge cities in China has played a positive role in promoting the total factor productivity of our cities.

In terms of the regulatory variables, the restructuring of industrial structure has a positive impact on total factor productivity, and it shows a significant feature in the region above 1%, indicating that with the continuous optimization of the urban development process, total factor productivity is improved, thus promoting the sustainable development of the urban economy. There is a significant positive correlation between infrastructure development and total factor productivity, and with the continuous improvement of infrastructure, it will improve total factor productivity, which in turn will promote the economic development of cities. Government intervention has a positive effect on total factor productivity and is around 1%, indicating that government intervention can increase total factor productivity, thus contributing to a win-win situation for both environmental and economic development. The estimated coefficient of population density has a negative value, which indicates that the increase in population density is not beneficial to economic development. This is because the population density is more than what the city can afford to develop, causing a weakening of the role of human capital, which in turn has an impact on the development of the city. The degree of impact of technological innovation on total factor productivity is significantly higher than 5%, indicating that technological innovation plays a pivotal role in improving total factor productivity in our national economy. The regression coefficients of economic agglomeration, human capital and the level of foreign openness are low, indicating that their effects on human capital, foreign openness and economic agglomeration are not significant. Among these three factors, human resources and economic agglomeration have become the main factors limiting the development of our cities. This indicates that in the process of promoting the “sponge city” in our country, we must strengthen human capital and economic agglomeration and gradually activate their positive effects on total factor productivity.

On this basis, the total factor productivity of China’s cities is further investigated and predicted. Model (2) is a regression based on incorporating technological innovation, industrial upgrading and infrastructure construction into model (1). The estimated coefficients of sponge city construction in models (1) and (2) are both positive and above 1%, which indicates that there is a large positive correlation between sponge city construction and total factor productivity. Technical efficiency refers to the difference between actual and optimal output, and technical progress refers to the change in the level of technology. Dividing the ml-ebm indicator into the technical efficiency indicator and the technical progress indicator, model (3) and model (4) further investigate the differences between the two indicators. It was found that the regression coefficient of technical efficiency as an explanatory variable was low and not significant. With technological progress as the explanatory variable, the results were positive and tested at 1%, indicating that technological progress is more closely related to sponge cities. This indicates that technological progress is an important factor in driving total factor productivity in cities.

### 4.2. Parallel Trend Test

When using the double difference method, the parallel trend test must be satisfied. Therefore, in this paper, the parallel trend test is required when double difference analysis is conducted. In other words, the economic development level of pilot sponge cities and non-pilot cities should be consistent before the construction of sponge cities, otherwise, the double difference method will have a greater or lesser impact on sponge cities. The following model was developed to test the common trend.
(3)$$ml\_ebm_{it}=\beta_{0}+v_{i}+\mu_{t}+\sum_{j\geq -7,j\neq 0}^{3}\beta_{j}C_{it}^{j}+\beta_{2}X_{it}+\varepsilon_{it}$$
where, $ml\_ebm_{it}$ denotes total factor productivity; *i* denotes individual, *t* denotes time, and *j* denotes the time relative to sponge city construction (*j* = −7, −6, …, 3); When *j* = −7, −6, …, 3, the $C_{it}^{j}$ This dummy variable takes the value of 1, otherwise it is 0. Since the first year of sponge city construction is taken as the base period in the parallel trend test in this paper, the dummy variable at *j* = 0 is removed from the model.

As can be seen from Table 4, in the years before the construction of sponge cities, the regression coefficients were not significant and the values were low, indicating that under the conditions of not including the sponge city pilot, there was no significant difference in the development trend of the cities, and the trend of urban economic development was more stable and flat. One year after the pilot sponge city, its regression coefficient reached a positive value, all have 1% statistical level of significance, and with the increase of time, the regression coefficient increased significantly, which indicates that the construction of sponge city has played a positive role in promoting the sustainable development of urban economy; the results confirm the results of the previous analysis of this paper and the parallel trend test, with high accuracy and reliability.

### 4.3. Robustness Tests

To better study the mechanism of the effect of sponge city construction on economic development, this paper tested its robustness and conducted regression analysis from two perspectives: first, based on the availability of data, the method described previously was used. Second, using Battese and Coelli’s single-level stochastic frontier method, GDP was measured as an input to output, labor, and capital, and its impact level was regressed to verify its impact on sustainable urban development.

From the estimated Equation (1) in Table 5, it can be seen that after the implementation of the sponge city, the primary lag coefficient of urban ecological benefit is 1.16, which is significant at 1%, indicating that the ecological level in the previous period has a positive effect on the current ecological effect. Also, the coefficients of the control variables used are consistent with the results of the previous analysis. From the estimated Equation (2), it can be seen that “indu” represents a primary economic lag, which still has a significant positive effect on the current level of urban sustainability, while the economic effect also has a positive effect on urban sustainability, and is highly significant at 5%. After the moderation of ecological and economic factors, the estimated urban sustainable development level in Equation (3) still remains significant at 5%, which indicates that the model has good predictive effect.

## 5. Conclusions and Recommendations

### 5.1. Conclusions

In the construction of the sponge city we need to change its original concept; on the basis of tradition, the concept of “sponge city” needs to be integrated into the construction and development of the city itself. At present, China’s sponge city development is facing technical and management problems. Sponge city construction will focus on solving these two problems, as well as managing the key technology breakthroughs, in order to allow the city to better adapt to the new building concept. Updating the concept of the city is an upgrade of the traditional concept of architecture, and the new concept will bring new, comprehensive benefits to the city. Regarding the exploration of the sponge city concept, before its implementation, it is necessary to conduct an in-depth analysis of relevant natural disasters caused by rainfall and to explore the various benefits that may arise from the construction of sponge cities. On this basis, this paper constructs a comprehensive benefit assessment index system from four aspects: environmental, economic, socioeconomic, and societal benefits, and conducts parallel tests and robust analysis to make the concept better validated. The analysis concludes that the emergence of sponge cities is an inevitable trend to achieve sustainable and green urban development, as well as an important measure to implement the idea of ecological civilization. Its research findings are as follows:(1)On the basis of sustainable development, ecological economy, government performance management, and system dynamics, a comprehensive performance assessment index for cities was established, one which includes urbanization rate, total annual runoff control rate, capital investment in sponge cities and surface water quality index of cities. The research indicators of this paper were screened from five dimensions: driving force, pressure, state, impact and response, and an empirical model that was constructed based on the actual needs of China.(2)In terms of trend prediction, the empirical results show that sponge city construction can also play a role in improving the ecological environment while pulling along economic development, and if the system remains unchanged, then future sponge city construction will show a stable development trend, greatly promoting the improvement of total factor productivity in the city, and the social and ecological benefits of the project will continue to improve, while its economic benefits will be, within a certain range, sufficiently stable, which shows that the development prospect of sponge city is very optimistic.(3)In terms of sensitivity prediction, indicators such as runoff pollution control volume, annual runoff, control rate, annual SS removal rate, policy release volume, investment amount of construction projects, and sewage treatment rate at the evaluation index level are more sensitive to the level of sustainable urban development performance, while indicators such as ecological shoreline percentage and natural lake maintenance rate are less sensitive to the project’s performance level, indicating that sensitive indicators such as runoff pollution control volume need to be paid attention to, and that professional technical guidance basis is issued accordingly to guide the later development.

### 5.2. Recommendations

#### 5.2.1. Harmonious Development of People and Nature

At present, the first task of the government is to change the development concept of “emphasizing the surface and neglecting the underground”. As an ecological project, the “sponge city” has laid a solid foundation for the construction of an “environment-friendly and resource-saving” society [69]. It is not only about the support and long-term carrying capacity of green areas and natural landscapes for cities, but also about the sustainability and connectivity of natural resources and drainage systems. Local governments should not only follow the central government conceptually, but also transform the construction of sponge cities into a new public service standard conceptually, in order to promote the harmonious development of the environment and the economy and to achieve harmony between people and nature, instead of blindly engaging in “performance projects” [70]. We should always adhere to the modernization concept of harmonious development of man and nature, keeping to the road of green ecological development, and correctly handling the friendly relationship between ecological environment and economic development.

#### 5.2.2. Focus on Training Professional and Technical Personnel

In contemporary social development, the issue of water ecological resources is a key issue, but it is also a complex issue, mainly because it involves a variety of factors. It is necessary to take into account both the rational use of water resources and flood control and drainage; in addition to the construction of green infrastructure, it also includes the construction of gray infrastructure; at the same time, attention must be paid to the prevention of water pollution, as well as landscape water. Therefore, the question is analyzed from multiple perspectives, resulting in the establishment of a multi-objective governance system, thus further realizing system governance, process control and source control. The establishment of an integrated rainwater management system will involve many scientific issues such as environmental concerns, municipality issues, meteorology, water conservancy, hydrology, etc., and will also involve many urban management departments such as water conservancy, urban construction, and environmental protection, so the communication and collaboration between various professions, as well as mutual support between various departments, is particularly important [71,72]. The construction of the sponge city requires the support and cooperation of various aspects such as environmental protection, water conservancy, landscape, municipalities, and roads. China’s major sponge city construction partners can build a “sponge city” strategic alliance platform, integrating construction enterprises, social capital, planning, design and consulting, academic research institutions, etc. together to set up a set of industry top-level design solutions and a complete industry chain with the platform operation mode of “technology + capital”. At the same time, it should also combine foreign advanced technology and introduce perfect water circulation system, WUSD, SUDS, BMP, LID and other technologies, so as to make a set of sponge city development plan that meets local reality according to local land use change, housing density, population density, local rainfall, geographical conditions, hydrological conditions and other factors. In addition, sponge city is a process of fine management, fine construction and fine design, so the management level of urban construction management should also be improved accordingly [73]. In conclusion, the development of sponge cities is a long way to go, and the country should increase investment in sponge cities, expand investment channels, increase government support, encourage scientific and technical workers to carry out research and development of new technologies as well as updating of technologies and equipment, and also vigorously train relevant technical personnel.

#### 5.2.3. Strengthening the Participation of Citizens in Sponge City Construction

Since China’s urban development model has a long history and has a great impact on the natural ecological environment of cities, it is very necessary to promote and popularize the concept of “sponge city” in urban construction. However, at present, the public awareness of sponge city is not high; there is a lack of awareness of the importance of water resources, so it is difficult to mobilize the whole society’s enthusiasm. In the process of promoting sponge cities, the government should strengthen publicity and adopt certain incentives and punishments to raise public awareness, and at the same time, the government should actively seek the support of mainstream media to enhance the publicity of sponge cities [74]. Sponge city is a new policy; most people are not familiar with the connotation of the sponge city. Accordingly, the government, through the media, can promote its goals and plans, so that non-governmental organizations and the public understand its meaning and significance, so as to better serve the public, and effectively deal with the problem of rainwater discharge. Sponge city construction is a reflection of the world trend: the national developmental needs of an ecological civilization project, the construction of a “green, livable” city as an inevitable requirement. Although the current era of China’s sponge city construction has achieved good results, compared with Europe and the United States and other developed countries’ sponge cities, there is still a big gap. Sponge city is a long-term, lasting project, and therefore, policymakers must strengthen the participation of all sectors of society in order to truly and effectively promote the construction of the sponge city.

## Figures and Tables

**Table 1 ijerph-20-01694-t001:** Sponge city construction performance evaluation system based on DPSIR-EES framework.

	Ecology	Economy	Social
Driving force D Pressure P			The number of
			Urbanization rate
			GDP
	Urban built-up area	Government revenue accounts for total revenue	Water resources per capita
	Per capita park green area	The proportion of	Water consumption for ecological environment
State S	Sewage	Government sponge city construction project	Total water
		Investment	
	Sponge area	Sponge city construction volume	Domestic water price
	Natural lake retention rate		
	Proportion of ecological shoreline		
	The rate		
	Sewage treatment rate		
Impact I	Standard rate of surface water quality	Sponge city capital input	Runoff pollution control
			Annual total runoff control rate
			Annual total SS removal rate
The response R	Project ecological benefit	Project economic benefit	Social benefit of project
			Related policy releases

**Table 2 ijerph-20-01694-t002:** Estimation results of the impact of sponge city construction on economic development and environmental pollution emission.

Variable	Model (1) GDP per Capita	Model (2) GDP per Capita	Model (3) Industrial Sulfur Dioxide Emissions	Model (4) Industrial Sulfur Dioxide Emissions
Datacity	0.691 ***	0.121 *	−0.408 ***	−0.339 ***
	(22.34)	(7.65)	(−12.26)	(9.44)
Indu		2.169 ***		0.854 ***
		(15.46)		(2.81)
Pop		−0.071		0.644 **
		(−0.53)		(2.13)
Skill		1.800 ***		−0.123
		(44.02)		(−1.37)
Inf		0.510 ***		0.261
		(6.72)		(1.58)
Hc		0.181 ***		−0.106 ***
		(13.53)		(−3.51)
Invest		0.078 ***		−0.060
		(4.98)		(−1.70)
Fdi		−0.061 ***		0.092 ***
		(−4.05)		(3.08)
Gov		0.108 ***		−0.045
		(6.22)		(−1.09)
Agg		0.718 ***		−0.105
		(16.48)		(−1.111)
City fixed effect	Yes	Yes	Yes	Yes
Year fixed effect	Yes	Yes	Yes	Yes
Constant term	10.465 ***	−8.117 ***	10.581 ***	7.423 ***
	(1288.33)	(−13.46)	(1230.27)	(5.62)
N	3703	3703	3703	3703
r^2^	0.139	0.815	0.053	0.068

Note: ***, ** and * are significant at 1%, 5% and 10% levels, respectively; Data in parentheses are t-statistics.

**Table 3 ijerph-20-01694-t003:** Estimation results of the impact of sponge city construction on green total factor productivity.

Variable	Model (1) ML-EBM	Model (2) ML-EBM	Model (3) EC-EBM	Model (4) TC-EBM
Datacity	0.043 ***	0.030 ***	−0.013	−0.032 ***
	(8.74)	(5.05)	(−0.40)	(3.57)
Indu		0.176 ***	0.130 *	0.046
		(4.87)	(2.04)	(0.76)
Pop		−0.175 ***	−0.177 ***	−0.031
		(−4.81)	(−2.79)	(−0.48)
Skill		0.035 **	−0.015	0.034
		(2.59)	(−0.35)	(1.62)
Inf		0.083 ***	−0.014	0.095 ***
		(3.92)	(−0.18)	(2.92)
Hc		−0.016	−0.021 *	0.016
		(1.62)	(−1.91)	(1.22)
Invest		−0.017	−0.012	−0.015
		(−1.84)	(−0.33)	(−0.78)
Fdi		0.015	−0.013	0.017
		(1.41)	(−0.55)	(1.37)
Gov		0.024 ***	0.028 **	−0.012
		(3.42)	(2.55)	(−0.31)
Agg		−0.013	−0.024	−0.012
		(−0.28)	(−0.79)	(−0.19)
City fixed effect	Yes	Yes	Yes	Yes
Year fixed effect	Yes	Yes	Yes	Yes
Constant term	1.012 ***	0.465 ***	0.958 ***	0.613 **
	(1035.09)	(3.15)	(3.68)	(2.68)
N	3703	3703	3703	3703
r^2^	0.032	0.076	0.016	0.031

Note: ***, ** and * are significant at 1%, 5% and 10% levels, respectively; Data in parentheses are t-statistics.

**Table 4 ijerph-20-01694-t004:** Parallel trends and dynamic effects.

Variables	ml-ebm	ml-ebm	ml-ebm
Pre7		−0.020	0.021
		(−1.60)	(1.70)
Pre6		−0.013	0.019
		(−0.41)	(1.13)
Pre5		−0.018	0.013
		(−0.97)	(0.34)
Pre4		−0.024	−0.019
		(−1.83)	(−1.16)
Pre3		−0.013	0.011
		(−0.40)	(0.16)
Pre2		−0.017	−0.017
		(−0.91)	(−0.83)
Pre1	0.012	0.016	0.013
	(0.21)	(0.75)	(0.43)
Post1	0.033 ***	0.039 ***	0.035 ***
	(3.85)	(3.65)	(3.19)
Post2	0.035 ***	0.041 ***	0.036 ***
	(3.58)	(3.61)	(2.09)
Post3	0.074 ***	0.080 ***	0.075 ***
	(6.48)	(6.25)	(5.93)
Control variables	Yes	No	Yes
Constant term	0.382 **	1.060 ***	0.352 *
	(2.59)	(113.49)	(2.39)
N	3703	3703	3703
r^2^	0.085	0.143	0.089

Note: ***, **, * indicate significant at the 1%, 5%, and 10% levels, respectively; data in parentheses are t-statistics.

**Table 5 ijerph-20-01694-t005:** Robustness test results.

	(1) Ecology	(2) Economy	(3) Social
datacity	1.160 *** (0.047)		1.031 *** (0.050)
indu		0.689 ** (0.270)	
pop			−0.221 ** (0.089)
skill			−0.013 *** (0.004)
inf	−0.096 ** (0.043)	0.020 ** (0.015)	0.017(0.016)
hc	0.384 *** (0.126)	0.194 ** (0.083)	0.045 ** (0.032)
invest	−0.096 ** (0.037)	−0.019(0.034)	0.016(0.026)
fdi	−0.021(0.017)	0.109 ** (0.051)	0.068 ** (0.032)
gov	0.014 *** (0.012)	0.032(0.372)	0.016(0.027)
agg	0.167 *** (0.664)	1.173(1.730)	0.008 ** (0.012)
Urban fixed effects	3.461 *** (0.963)	4.632 *** (0.741)	0.384 ** (0.661)
Year fixed effects	2.456 *** (0.652)	2.963 ** (0.461)	0.731 *** (0.963)
Constant term	3.014 *** (0.521)	3.496 *** (0.783)	0.448 * (0.675)
N	0.110	0.187	0.739
r^2^	0.281	0.861	0.251

Note: ***, **, * indicate significant at the 1%, 5%, and 10% levels, respectively.

## Data Availability

The data are not publicly available due to privacy restrictions.

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
