# Peer review of "Sponge City Construction and Urban Economic Sustainable Development: An Ecological Philosophical Perspective"

_ijerph, 2023, doi:10.3390/ijerph20031694_

Round 1
Reviewer 1 Report (Previous Reviewer 2)
Thank authors for revisions. I agree with the publication. I believe that the language of this manuscript can be improved.
Author Response
- English language and style are fine/minor spell check required.
Response: Thank you for your valuable suggestions. The author has carefully checked the writing of the article and corrected some of the poorly expressed elements. This modification is marked in red. See line number:58-62,70-77,88-92,101-108,118-119,138-143,194-199;
- The results can be improved.
Response: Thank you for your valuable suggestions. The authors fine-tuned the conclusion section based on the results of the empirical analysis. This modification is marked in purple. See line number:524-533;
This manuscript is a resubmission of an earlier submission. The following is a list of the peer review reports and author responses from that submission.
Round 1
Reviewer 1 Report
This manuscript presents an interesting work concerning the relationship between the sponge city construction and the sustainable economic development of a city. The ecological philosophy issues is addressed. The research fits the scope of this journal well. However, there are some questions needed to be answered before the accept of this manuscript. My comments are listed as below:
1. The abstract needs to be carefully checked and rewritten. The authors should emphasize on the significance, main findings, contribution and innovation of your work.
2. The introduction section is hard to understand. In general, in the introduction section, the authors could introduce the research background, literature review, limitations of current research (or knowledge gap) and the significance of your work, followed by the structure of the manuscript. I suggest you reorganize and rewritten this section.
3. In section 2 “research hypothesis”, you have section 2.1 but no section 2.2, please carefully check and avoid long paragraphs, which is hard to read. Similarly, research hypothesis should explain clear the theoretical foundation of the research hypothesis but instead of a literature review. In other words, authors should state clearly why this hypothesis should be made and in what sense it is reasonable.
4. The ecological and economic variables are both represented by E. Please differentiate them.
5. The data source section needs to introduce what kind of data is used and how to obtain and process the data, instead of variable selection.
6. The analysis of the results are insufficient. For instance, table 4 lists the results of parallel trends and dynamic effects, what does the results mean and why? What are the empirical meaning of the result.
7. The manuscript needs to be polished up. The format of the reference should be checked carefully.
Reviewer 2 Report
1. In the Introduction, this study not only provides the theoretical background, but also clarifies the literature contribution. Although authors believe that the new concept has been put forward, I do not know what the concept is? Meanwhile, how the importance of new concept is?
2. I hold that the Literature review needs to be written in this study. I can not find what the literature contribution?
3. In the Research-area overview, data, methods, this study should need to carefully present the literature source of each indicator. Meanwhile, each formula should be given the serial number.
4. In the Result, I hold that author doesn’t give explanation for the results, this study only presents the result according to the Table. More seriously, there is a lack of both robustness test and endogeneity test. Therefore, I suspect that the results are unbelievable.
5. I strongly suggest that author write the Discussion. In Discussion, author should make comparison between these conclusions and the extant literature to reveal the innovation. More importantly, I don’t know what are the literature contributions of this study?
6. In the Reference, authors quote too much Chinese literature, indicating that author don’t adequately know the research progress of this topic. Meanwhile, I strongly suggest using a professional and native English academic writing proofreading service to enhance the readability and quality of the language.